# Three-Dimensional Printed Carbon Black/PDMS Composite Flexible Strain Sensor for Human Motion Monitoring

**DOI:** 10.3390/mi13081247

**Published:** 2022-08-02

**Authors:** Haishan Lian, Ming Xue, Kanglin Ma, Deyun Mo, Lei Wang, Zaifu Cui, Xiaojun Chen

**Affiliations:** 1School of Mechanical and Electronic Engineering, Lingnan Normal University, Zhanjiang 524048, China; lianhs@lingnan.edu.cn (H.L.); mkl447314@163.com (K.M.); dewin_mo@163.com (D.M.); zaifucui@lingnan.edu.cn (Z.C.); 2School of Intelligent Manufacturing, Dongguan Technician College, Dongguan 523112, China; sn2006@126.com

**Keywords:** direct-write 3D printing, CB/PDMS composites, strain sensors, motion monitoring

## Abstract

High-performance flexible strain sensors with a low cost, simple structure, and large-scale fabrication methods have a high demand in soft robotics, wearable devices, and health monitoring. Here, a direct-ink-writing-based 3D printing method, which fabricates structural layers in an efficient, layered manner, was developed to fabricate a stretchable and flexible strain sensor composed of carbon black/silicone elastomer (CB/PDMS) composites as the strain-sensing elements and electrodes. As the sensing element, the CB/PDMS composite had a sensitivity of 5.696 in the linear strain detection range of 0 to 60%, with good stability and low hysteresis. The flexible strain sensor demonstrates potential in monitoring various human motions, including large deformation motions of the human body, and muscle motions with facial micro-expressions.

## 1. Introduction

Owing to their potential applications in wearable electronics [1,2], electronic skin [3,4], soft robots [5], and health monitoring systems [6], flexible strain sensors have attracted a significant amount of attention in recent years. As an important intermediate for collecting external mechanical signals, flexible strain sensors play a central role and are characterized by flexibility, compliance, and robustness [7]. Among various flexible strain sensors, piezoelectric, piezoresistive, triboelectric, and capacitive strain sensors have the advantages of a simple structure, easy signal acquisition, and high sensitivity and have attracted many researchers [8,9,10]. For example, a strain sensor based on the piezoresistive effect has been fabricated by dynamic crosslinking of conductive nanocomposites, with a sensitivity as high as 46, which can detect various deformations such as stretching, bending, and torsion [11]. In addition, PDMS mixed conductive fillers were used to fabricate a flexible strain sensor with a sensitivity of 4.5 in the linear strain range of 0–20% [12]. Although a flexible strain sensor with high stretchability and sensitivity could be directly realized by filling conductive nanomaterials into elastomers, a low-cost and large-scale fabrication method is still needed for widespread application of the device.

In the existing literature, many flexible fabrication methods have been proposed to fabricate flexible strain sensors, such as photolithography [13], transfer printing [14,15], spray coating [16], and laser direct writing [17]. These methods often have complicated operation steps, low efficiency, and a high cost. Surprisingly, 3D printing is widely used to fabricate flexible devices due to its advantages of high precision, simplicity, and rapid prototyping. For example, Cheng et al. fabricated a flexible strain sensor of reduced graphene oxide/elastomer resin (RGO/ER) composites using DLP-based 3D printing technology. The RGO/ER composite exhibited a sensitivity of 6.723 within a linear detection range of 40% and good stability over 10,000 stretching–relaxation cycles [18]. Kim et al. used multi-walled carbon nanotubes (MWCNTs) and polylactic acid (PLA) composite conductive polymers to fabricate flexible pressure sensors for multidirectional force detection by FDM printing, realizing the distinction between different human actions and grasping actions [19]. By controlling the solid–liquid phase transition of paraffin wax, Wang et al. achieved the fabrication of 3D soft silicone mesh devices using direct ink writing 3D printing technology. The printed structure treated with conductive carbon nanotubes can be used as a flexible pressure sensor to detect wrist pulses [20]. Compared with DLP and FDM, direct ink 3D printing technology directly extrudes composite materials through nozzles to manufacture micro-nanostructures. It can print a wide variety of materials with a low cost and satisfactory production efficiency, thus gaining continued attention in the fabrication of various flexible strain sensors.

Silicone elastomers are often mixed with conductive materials (silver nanowires [21], carbon nanotubes [22], graphene [23], liquid metals [24]) to fabricate stretchable strain sensors. However, the low viscosity and modulus of most silicone rubber precursors make it difficult to directly use them for direct ink writing 3D printing. One method is to change the viscosity of the silica gel by adding fumed silica as a thixotropic agent, enabling the printing of the silicone rubber precursor. However, the addition of silica significantly reduces the elasticity of the silicone rubber, and crosslinking with the silicone rubber also causes the material to harden over time [25]. Furthermore, a low-cost conductive nanoparticle (carbon black) as a conductive filler in an elastomer matrix has good wettability and high compliance, which is particularly suitable for layer-by-layer 3D printing in the fabrication of flexible devices [26].

Here, a carbon black (CB)/silicone rubber (PDMS) composite ink material was developed for direct ink writing 3D printing. The rheological properties of PDMS precursors are regulated by carbon black powder to achieve layer-by-layer 3D printing fabrication, which endows composites with high electrical properties and improves the performance of materials in flexible sensors. In this paper, a CB-filled PDMS composite-based flexible strain sensor with a sensitivity of 5.696 (linear range of 0–60%) was fabricated by direct ink 3D printing technology. As an application, it demonstrates the potential of strain sensors in monitoring human motion.

## 2. Materials and Methods

### 2.1. Materials

PDMS (Sylgard 184) and curing agents were purchased from Dow Corning (Midland, MI, USA). Carbon black (EC600JD) was purchased from Suzhou Shinerno Technology Co., Ltd., Suzhou, China. Anhydrous ethanol was of analytical grade and was purchased from Shanghai Titan Technology Co., Ltd., Shanghai, China.

### 2.2. Preparation of CB/PDMS Conductive Composites

The preparation process of carbon black/PDMS conductive composites is shown in Appendix A. The detailed steps were as follows: Granular carbon black was ground to a powder and placed in a plastic cup for later use. Then, an appropriate amount of absolute ethanol was added to the plastic cup containing the carbon black powder. An ultrasonic cleaner (CR-040S, purchased from Shenzhen Chunlin Cleaning Equipment Co., Ltd., Shenzhen, China) was used to disperse the carbon black powder in anhydrous ethanol with an ultrasonic power of 240 w. In the meantime, an auxiliary electric stirrer (LC-OES-60SH, purchased from Shanghai Lichen Instrument Technology Co., Ltd., Shanghai, China) was stirred at a stirring speed of 1500 r/min for 30 min in order to evenly distribute the carbon black powder in anhydrous ethanol. Then, an appropriate amount of PDMS solution was added to the carbon black solution for 30 min at an ultrasonic power of 240 W and a stirring speed of 1500 r/min. Next, the temperature of the ultrasonic cleaning machine was adjusted to 60 °C, and stirring and ultrasonic treatment were continued under the same conditions until the absolute ethanol was completely volatilized. Finally, the curing agent of PDMS (10:1) was added, and stirring was performed at a speed of 1500 r/min for 10 min, until the CB/PDMS solution and curing agent were uniformly dispersed. The preparation of the CB/PDMS conductive composite was then completed.

### 2.3. Three-Dimensional Printed Flexible Strain Sensors

The flexible strain sensor with the CB/PDMS composite material was realized by a self-made 3D printer, which includes a moving axis in the x-y-z direction, a nozzle, a pneumatic extrusion device, and a control system. First, the prototype was modeled using SolidWorks software and converted into an STL model that could be read and written by a printer. Then, slicing was performed on slicing software and uploaded to the printer’s database. Then, the morphology of the 3D printed structure was adjusted by setting the process parameters such as the printing layer height, printing speed, and material extrusion speed. The CB/PDMS conductive composite was poured into a syringe barrel at room temperature, and the 3D structure was printed by air pressure extrusion.

### 2.4. Characterization and Testing

Characterization: The morphology of the printed samples was observed with a field emission scanning electron microscope (XL-30, PHILIPS Instrument Co., Ltd., Eindhoven, The Netherlands) and an optical microscope (CSW-H2KACL, Shenzhen Kishiwei Optical Instrument Co., Ltd., Shenzhen, China). Field test pictures were taken with mobile phones.

Electromechanical performance test: Tensile and compressive tests were carried out with a digital push–pull gauge (HP-100, Yueqing Aidelberg Instrument Co., Ltd., Wenzhou, China). The conductive properties of the CB/PDMS composites were demonstrated by lighting LED lights with a DC regulated power supply (RXN-305D, Shenzhen Zhaoxin Electronic Instrument Equipment Co., Ltd., Shenzhen, China). The electrical properties of the composites were recorded with a digital multi-meter (VC980E, Shenzhen Victory Yisheng Technology Co., Ltd., Shenzhen, China). GF can be defined as
GF = (ΔR/R0)/ε = ((R − R0)/R0)/ε 

In the formula, ΔR is the change in resistance before and after straining, R0 is the initial resistance, R is the resistance after applying strain, and ε is the applied strain.

## 3. Results

### 3.1. Three-Dimensional Printed Strain Sensors with CB/PDMS Composites

The CB/PDMS-based strain sensor was fabricated using a homemade 3D printer. The principle of direct writing 3D printing is shown in Figure 1a. The material in the barrel is pushed by the air pressure. The printing ink is continuously squeezed out to solidify the stack and realize the construction of the 3D structure (Figure 1b, and Appendix A). By using different ink materials, various structures with different properties can be printed. Figure 1c–e show photographs of pentagrams, 3D multilayer grid structures, and 3D honeycomb structures printed by direct-write printers.

Furthermore, the 3D printed strain sensors are highly flexible and robust, allowing them to be stretched after crimping (Figure 1f). Due to the elasticity of CB/PDMS, this composite-based mesh structure can be compressed and recovered (Figure 1g). It can withstand large compression and quickly return to the initial state after the pressure is removed, exhibiting excellent mechanical elasticity. In addition, the structure of the 3D printed strain sensor is tough enough. As shown in Figure 1h, a strip with a width of 6 mm and a thickness of 1.5 mm can carry a weight of 200 g without breaking.

### 3.2. Properties of CB/PDMS Composites

Carbon black is a low-cost conductive nanoparticle. As a filler in a PDMS elastomer matrix, it can improve the mechanical strength, flexibility, and stretchability of composites. Carbon black was added to PDMS to construct a conductive network for the fabrication of flexible strain sensors. Figure 2a shows an SEM image of the pure PDMS surface without carbon black addition. The printed structure has a smooth surface without impurities. The carbon black particles are irregular round spheres with a diameter of 30–50 nm. There is agglomeration between particles rather than a dispersed state of individual particles (Figure 2b). After the addition of carbon black, the surface of the entire CB/PDMS composite appears rough (Figure 2c). The surface-layered carbon black indicates that the carbon black particles are dispersed in the PDMS, which will promote the formation of the graphene conductive network. The cross-section of the 3D printed layered structure shows that the carbon black is uniformly mixed into the PDMS (Figure 2d).

Carbon black particles are considered spherical in shape. There is point contact between particles. When subjected to external force, a small elongation can lead to a rapid reduction in the contact conductive channels formed by spherical particles in the material. Its volume resistivity responds rapidly to particle displacement, so it has high sensitivity. The incorporation of carbon black into PDMS forms a conductive network inside the composite for strain sensor applications. The changes in the distribution state of carbon black particles in the conductive composites during the stretching process are shown in Figure 2e. The macroscopic resistance of the carbon black conductive network is mainly composed of the contact resistance and tunneling resistance between the individual carbon black particles [27]. At relatively small strains, most of the carbon black particles remain attached. However, the contact resistance increases because the strain induces slippage of the carbon black, and thus the overlapping area in the carbon black decreases [28]. A further increase in strain may lead to a decrease in the number of overlapping carbon black particles, and thus the tunneling resistance between carbon black particles dominates.

In our experiments, the conductive properties of CB/PDMS composites with doping mass ratios of 4 wt%, 5 wt%, 6 wt%, and 8 wt% were compared. When the mass ratio was 4 wt% and 5 wt%, the resistances of the composites were 717 kΩ and 63 kΩ, respectively (Figure 3a,b). When the mass ratio was 6 wt%, the resistance of the composite was only 6 kΩ. Compared with the mass ratio of 4 wt%, this is two orders of magnitude lower (Figure 3c). The sample with a small carbon black content had a large resistance value, which is not conducive to the test of weak electrical signals. In addition, the incorporation of carbon black above 8 wt% greatly increases the viscosity of the composite material, and it is difficult to form connections between polymers, resulting in difficult printing. Therefore, considering the conductivity and printability, the 6 wt% CB/PDMS composite was selected for subsequent experiments.

The electrical properties of 3D printed CB/PDMS composites were investigated. As shown in Figure 3d, the LED light changed from bright to dim after the sample was stretched from the initial state. After stretching, the contact resistance increased, causing the LED lights to dim. After shearing, the sample was manually connected, the internal conductive network was reconnected, and the LED lights came back on (Figure 3e).

### 3.3. Electromechanical Properties of CB/PDMS Composites

The mechanical properties of 4 wt% and 6 wt% CB/PDMS composites and PDMS in compression were compared using the uniaxial compression test method. When the compression deformation of the sample was 50%, the stress–strain curve was obtained (Figure 4a). In the strain range of 0–50%, the stress and strain of different samples varied approximately linearly. The compressive modulus of the sample was obtained by fitting (Figure 4b). Compared with the pure PDMS material, as the concentration of carbon black increased, the corresponding stress under the same strain was larger, and the deformation was smaller. Therefore, doping carbon black into PDMS materials can alter their mechanical properties.

Figure 5a shows the tensile stress–strain curves of the 3D printed CB/PDMS composites. The tensile stress increased with increasing carbon black content. Compared with the pure PDMS, the elongation at break of the CB/PDMS composite decreased, while the elastic modulus increased slightly. The stress with a content of 6 wt% CB/PDMS was the largest among the three samples at the same strain. The possible reason is the mechanical reinforcement of the carbon black filler. CB/PDMS composites with a low elastic modulus have good flexibility for flexible strain sensors. To evaluate the stability of the flexible strain sensor, the stress–strain curve at 50% deformation over five cycles was investigated (Figure 5b). In the loaded state, the stress of the 3D printed sensor gradually increased with the strain. In the absence of tension, the stress returned to the initial state. This indicates that the 3D printed strain sensor has good stability.

Figure 5c shows the current–voltage diagrams of CB/PDMS composites with different widths at the same thickness (1.5 mm). As the width increases, the conductive network strengthens, resulting in an increase in current flow. The current–voltage curves are strictly in accordance with Ohm’s law. Figure 5d shows the relative resistance change (ΔR/R0)–strain curves of the CB/PDMS composites. The CB/PDMS composite can be stretched up to 60% with an increase in ΔR/R0 and good linearity. The increase in ΔR/R0 is attributed to the reduction in the conductive network formed by carbon black during stretching. The measurement factor (GF) is often used to evaluate the sensitivity of a sensor. The GF of the CB/PDMS sensor reached 5.696 within 60% strain.

We also tested the response of the sensor under different bending angles. Figure 5e shows that the resistance change exhibited an approximately linear relationship as the angle increased. When the CB/PDMS composite material changes from the tensile state to no tensile force, its electrical resistance cannot return to its original state. Figure 5f shows the hysteresis behavior of CB/PDMS composites under different strains. This phenomenon can be explained by the following two points: (1) the 3D printed CB/PDMS composite is an elastomer with inherent hysteresis [29]; (2) the carbon black conductive network decreases and increases during stretching. During the release process, a certain delay is expected in the reconstruction of the conductive network [30].

### 3.4. Application of 3D Printed Flexible Strain Sensor

A flexible strain sensor using the CB/PDMS composite as electrodes was printed for real-time monitoring of human motion (Figure 6a). Three-dimensional printed flexible strain sensors are mounted on fingers, wrists, and faces for human motion monitoring. The electrical signal of finger bending is recorded in Figure 6b. The resistance increased with gradual flexion of the joint, returning to its initial value after the joint returned to its original position. When the finger was bent 45 degrees, the average peak value of the resistance was 27 kΩ. When the finger was bent 90 degrees, the average peak value of the resistance was 34 kΩ, and its resistance change rate ΔR/R0 increased from 0.65 to 1.07. Therefore, this sensor can be integrated into a robotic glove to distinguish different degrees of finger motion variation. We also attached the sensor to the wrist joint, demonstrating its ability to detect continuous motion of the wrist joint (Figure 6c). The average value of the resistance peaks for three consecutive wrist inward bends was 42.6 kΩ, with a resistance change rate of 0.72. The bending motion of the wrist is displayed through the change in the resistance value. Therefore, in the future, sensors can be integrated into wristbands, watches, and other objects to detect the movement of the wrist in real time.

The application of the sensor in finger and wrist joints demonstrates its ability to detect large deformation of the human body. We also tested the changes in the sensor’s electrical signal through mouth opening and closing motions by fitting the sensor to a human cheek. As shown in Figure 6d, when the tester opened his mouth, the resistance increased; the average value of the resistance peak was 18.67 kΩ, and the resistance change rate was 0.13. These results show that the sensor can monitor tiny facial skin muscle movements of the human body. Therefore, strain sensors based on CB/PDMS composites have potential prospects in future applications such as human wearable devices and human motion monitoring.

## 4. Conclusions

In this work, we developed a flexible strain sensor based on direct ink 3D printed CB/PDMS composites. The device uses CB/PDMS composites as sensing elements and CB/PDMS with a multilayer grid structure as electrodes. The CB/PDMS sensing element had a sensitivity of 5.696 in a linear detection range of 60%, and low hysteresis. In addition, the application of 3D printed strain sensors in monitoring wrist movement and facial expression was also investigated. The results indicate that the sensor has broad application prospects in the field of intelligent detection.

## Figures and Tables

**Figure 1 micromachines-13-01247-f001:**
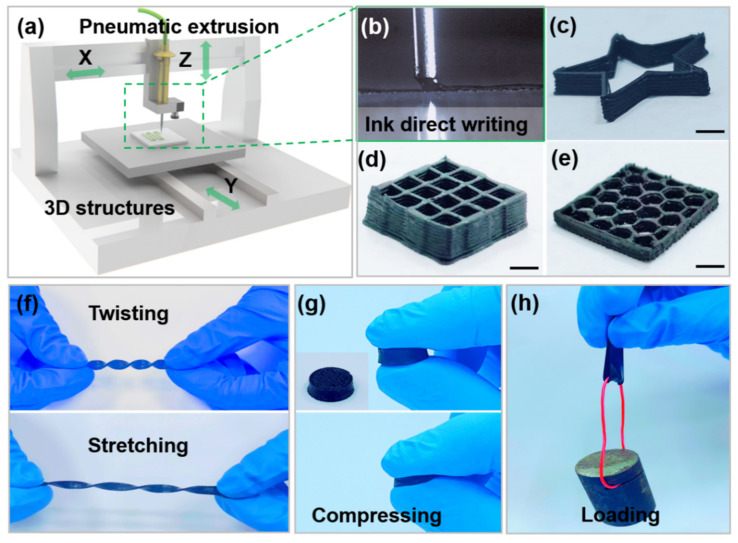
(**a**) Schematic diagram of direct ink 3D printing principle; (**b**) 3D printing process of direct ink writing; (**c**) 3D printed pentagram; (**d**) multilayer grid structure; (**e**) honeycomb structure; (**f**) twisting and stretching; (**g**) compression deformation and recovery; (**h**) a 200 g sample load.

**Figure 2 micromachines-13-01247-f002:**
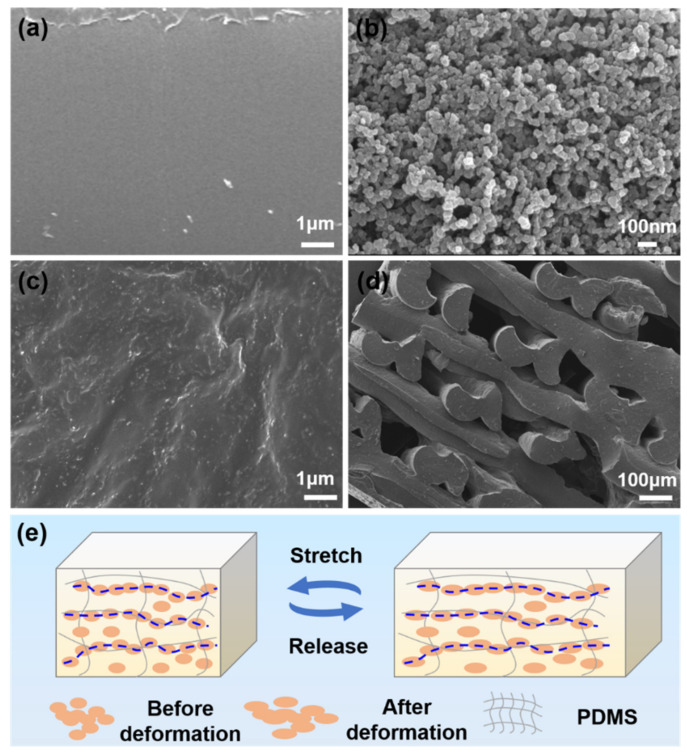
(**a**) Pure PDMS; (**b**) SEM image of carbon black particles; (**c**) CB/PDMS composite; (**d**) mesh structure of 3D printed composite; (**e**) conductive path change of CB/PDMS composite during mechanical deformation.

**Figure 3 micromachines-13-01247-f003:**
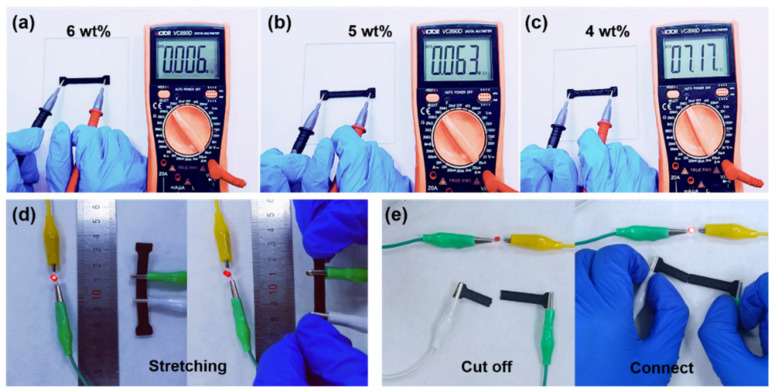
Electrical testing of 3D printed CB/PDMS composites. (**a**–**c**) Resistance measurement of composites with different ratios; (**d**) LED light brightness after stretching; (**e**) LED light for manual connection after cutting.

**Figure 4 micromachines-13-01247-f004:**
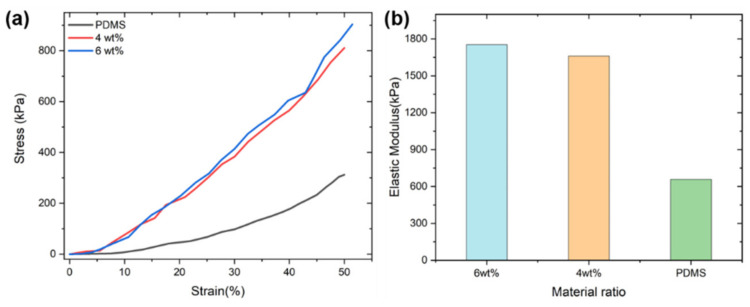
(**a**) ΔR/R0 as a function of applied compressive strains within 50% with different contents of CB particles; (**b**) compressive elastic modulus of the sample.

**Figure 5 micromachines-13-01247-f005:**
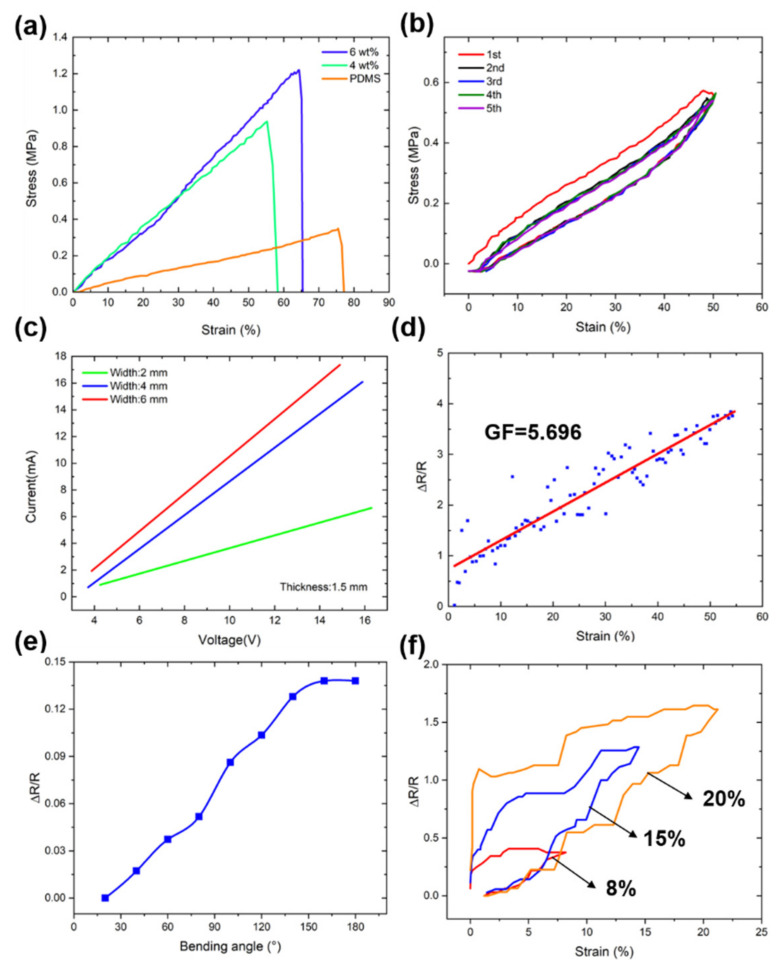
(**a**) Stress–strain curves of the CB/PDMS composites and PDMS; (**b**) cyclic stress–strain curves of five experiments; (**c**) current–voltage curves of the CB/PDMS composite with different widths at a thickness of 1.5 mm; (**d**) relative resistance change–strain curve of the CB/PDMS composite; (**e**) the relationship between the bending angle and resistance change; (**f**) electrical hysteresis curves of the CB/PDMS composite at strains of 8%, 15%, and 20%.

**Figure 6 micromachines-13-01247-f006:**
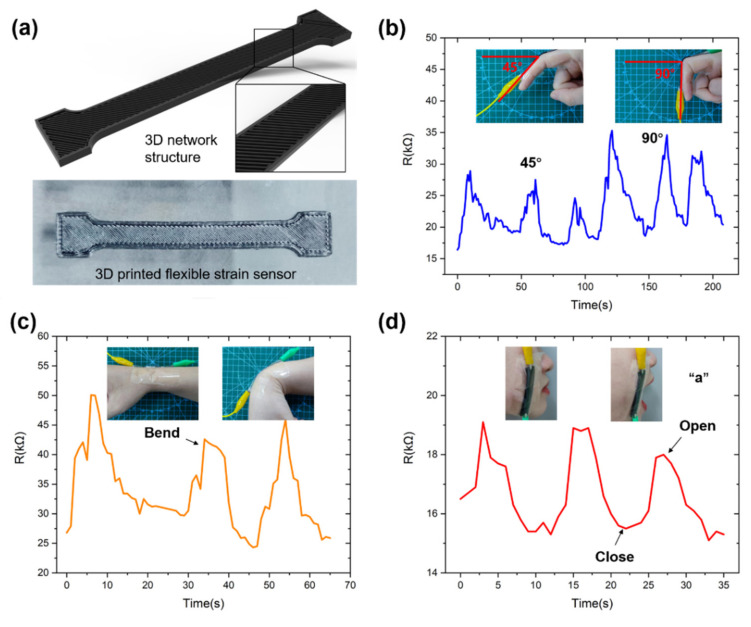
Application of the flexible strain sensor. (**a**) Three-dimensional printed flexible strain sensor. (**b**) Electrical responses of fingers at different bending angles. (**c**) Electrical response of the sensor attached to wrist flexion. (**d**) Mounting sensors to the face to detect facial muscle movements.

## Data Availability

Not applicable.

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
