# Peer review of "Three-Dimensional Printed Carbon Black/PDMS Composite Flexible Strain Sensor for Human Motion Monitoring"

_micromachines, 2022, doi:10.3390/mi13081247_

Round 1

Reviewer 1 Report

This paper presents a wearable sensor fabricated by carbon black and PDMS composite. This composite is further 3D printed for various designs for measuring various human motions. Here are some minor questions:

1)      What are the requirements for measuring motions? Requirement of Maximum strain and minimum gauge factor range can be provided from other references to properly measure human motions.

2)      What is the mechanism of the self-healing?

3)      How did the structure cure after printing? How do authors manipulate to print 3D structures? Since if a new layer is printed above an uncured layer, there could be issues in the morphology.  

Reviewer 2 Report

This paper describes the development of a 3D-printable material that can produce a conductive material on a bulk scale.

If additional content is described, it is judged that it can be published as a Micromachines article.

1. The description of the process of making Cunductive PDMS composite is insufficient. It should be written more specifically.

2. There is no analysis on the size or physical properties of the added carbon black particles. At least the SEM image should be added.

3. The schematic in which the conductive path in Figure 2(d) is is unclear to understand. You need to draw more detail to help readers understand.

4. There is no error bar in the graph. With the current graph, it can only be understood that the thesis was written as the result of one experiment.
